

# Development of the Community Active Sensor Module (CASM): Forward Simulation

B. T. Johnson[1,2] and S. A. Boukabara[2]

[1]Atmospheric and Environmental Research Corporation, Lexington, MA, USA
[2]National Oceanic and Atmospheric Administration (NOAA)

*Correspondence to:* B. T. Johnson (benjamin.t.johnson@noaa.gov)

**Abstract.** Modern data assimilation frameworks require sophisticated physical and radiative models to guide assimilation and interpretation of satellite-based observations. To date, satellite-based infrared and passive microwave radiances, in various scenarios, are being assimilated operationally at multiple centers around the world (e.g., ECMWF, NOAA), however precipitating/cloudy radiances assimilation is still under development for most observation streams. Additionally, with the advent of space-based precipitation radars (e.g., TRMM, GPM, CloudSat), active microwave scatterometers (e.g., RapidScat), and radar altimeters (e.g., JASON), interest in directly assimilating satellite-based active microwave observations is increasing. This paper describes the development of the Community Active Sensor Module (CASM), which is designed to simulate active microwave sensor observations, consistent with current and future sensors. This paper presents the forward modeling component of CASM, providing a model description, key physical elements, and sensitivity to the various inputs and implicit / explicit assumptions. CASM is also evaluated against the the Global Precipitation Measurement Mission Dual-Frequency Precipitation Radar (GPM DPR) observations in both a targeted case study and a global, year-long analysis.

## 1 Introduction

Satellite data assimilation requires algorithms to properly ingest, process, and interpret a wide range of satellite-based observations. With the advent of space-based precipitation radars (e.g., TRMM, GPM, CloudSat) and active microwave scatterometers (e.g., RapidScat, ASCAT), interest in directly assimilating satellite-based active microwave observations is rapidly increasing. The present research describes the ongoing development of Community Active Sensor Module (CASM): a framework for efficiently simulating active microwave observations using shared libraries from a suitable radiative transfer platform. The present work uses libraries from the Community Radiative Transfer Model (CRTM) to compute atmospheric absorption, scattering, and surface reflection properties (Kleespies, et al., 2004).

When provided with the appropriate physical description of the surface and atmosphere, CASM is designed to provide a standalone unified framework to simulate the surface and atmospheric response to actively emitted microwave radiation, with a specific focus on radar, altimeter, and scatterometer simulations. When completed, CASM will provide simulated atmospheric reflectivities, path-integrated attenuation, and surface normalized radar cross-section in all-weather conditions for any active



microwave sensor platform. The present paper describes the use of CASM to produce a forward simulation of nadir-viewing active radar reflectivities and path-integrated attenuation in a precipitating cloud scene.

The next development step for CASM is to compute the tangent linear and adjoint models for the forward operator, consistent with the operation of CRTM – this will be the subject of a future publication. The computation of the Jacobians of active sensor observables will allow for accurate derivation of atmospheric and surface parameters, to which the active sensors are sensitive (e.g., wind, wave height, cloud and precipitation profiles, etc.) in a variational data assimilation / 1D-VAR framework, such as Multi-Instrument Inversion and Data Assimilation Preprocessing System (MIIDAP) (Garrett et al., 2015). MIIDAPS provides a universal quality control algorithm for all satellite data observations and retrieval algorithm. The integration of CASM within MIIDAPS (or other similar 1D-VAR frameworks) provides a self-contained active forward operator for use in applications where the assimilation of active sensor observations is desired and the adjoint and tangent-linear models are also needed.

For sensors that have both active and passive capabilities, CASM, when combined CRTM, provides simultaneous active and passive simulation capabilities – useful for single-sensor active-passive observations, sensor quality control and cross-calibration, etc.) In the present version, only 1-D vertical profiles of reflectivity and path-integrated attenuation are provided. Ground-based radars are not explicitly considered here, but may be adapted at a future time. In the present version, there is no explicit treatment of slant-path/off-nadir radar reflectivities or radar multiple scattering enhancements (Battaglia et al., 2010).

A few prior researchers have created models for simulating satellite radar reflectivities and attenuation, such as Quickbeam (Haynes, et al., 2007), and some existing research models have been modified to compute radar reflectivities such as is described in Johnson et al. (2012) and Johnson et al. (2016). Di Michele et al. (2012, 2014) extended the "ZmVar" model for use as a lidar / radar forward operator in data assimilation at ECMWF, for use in 1D-VAR + 4D-VAR data assimilation frameworks. The present work is similarly designed with operational capabilities in mind, and as such, is designed with computational efficiency in mind.

## 2 CASM Technical Description

As introduced previously, The Community Active Sensor Module (CASM) is designed to simulate the active microwave response to the atmospheric and surface properties under all weather conditions. It accomplishes this using the Community Radiative Transfer Model (CRTM) libraries to provide necessary physical, scattering, and absorption properties of the atmosphere and surface. Figure 1 depicts the target design, inputs, processes, and outputs for CASM.

A separate reference model is used for comparison with CASM simulations, the details of which can be found in Johnson et al. (2012). The following sections describe the CRTM libraries and the standalone model codes used for the computation of radar reflectivities and path-integrated attenuation.



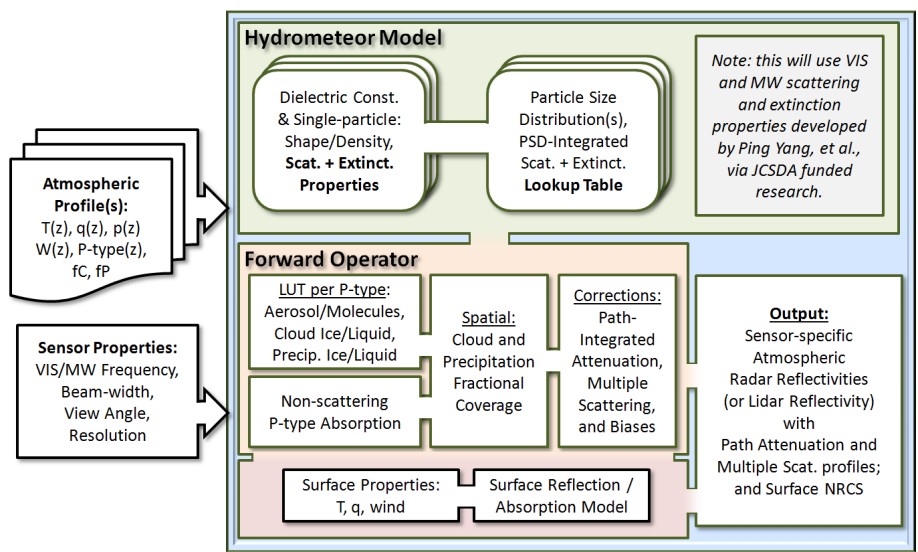

T = temperature, q = specific humidity, p = pressure, W = hydrometeor water content by type, P-type = hydrometeor type(s), fC = cloud fraction, fP = precipitation fraction

**Figure 1.** CASM design diagram, targeting the final form of the model.

## 2.1 Physical Properties of the Reference Model

The hydrometeor model, described in Johnson et al. (2012), is capable of distributing the user-prescribed water content using a four-parameter modified-gamma particle-size distribution (PSD) (Deirmendjian, 1969). For this study, we have adopted a simplified form of the modified-gamma distribution, namely the exponential PSD:

$$N(D) = N_0 \exp\left(-3.67 \frac{D}{D_0}\right), \tag{1}$$

where $D$ is the liquid-equivalent diameter, $N_0$ is the intercept parameter, and $D_0$ is the median diameter of the PSD. $D_0$ is related to the slope ($\Lambda$) of the exponential PSD by $\Lambda = 3.67/D_0$ (Ulbrich, 1983).

To maintain consistency with the CRTM default scattering properties, a spherical particle shape is used for both solid and liquid phase hydrometeors. However, although not presented here, the hydrometeor model also allows realistically shaped particles with extinction and scattering properties generated from the discrete dipole approximation (DDA) (see Johnson et al. (2016)). For spherical particles, we've selected the particle densities for snow, graupel, and hail to be consistent with what is assumed by default in CRTM. For snow, the density is $0.1 \, \mathrm{g\,cm^{-3}}$, for graupel it is $0.4 \, \mathrm{g\,cm^{-3}}$, and $0.9 \, \mathrm{g\,cm^{-3}}$ for hail.

The choice of density maps to a frequency-dependent average dielectric constant, according to either of two models for the dielectric constant for pure ice and one of three models for the dielectric constant of a mixture of ice and air (Johnson, 2007). We then calculate radiative cross sections for individual particles using standard Mie theory (Mie, 1908), and then integrate



over the specified PSD to obtain bulk radiative properties given an ensemble of particles. In the default CRTM scattering table, used here, there is no specified temperature dependence for the solid particles types.

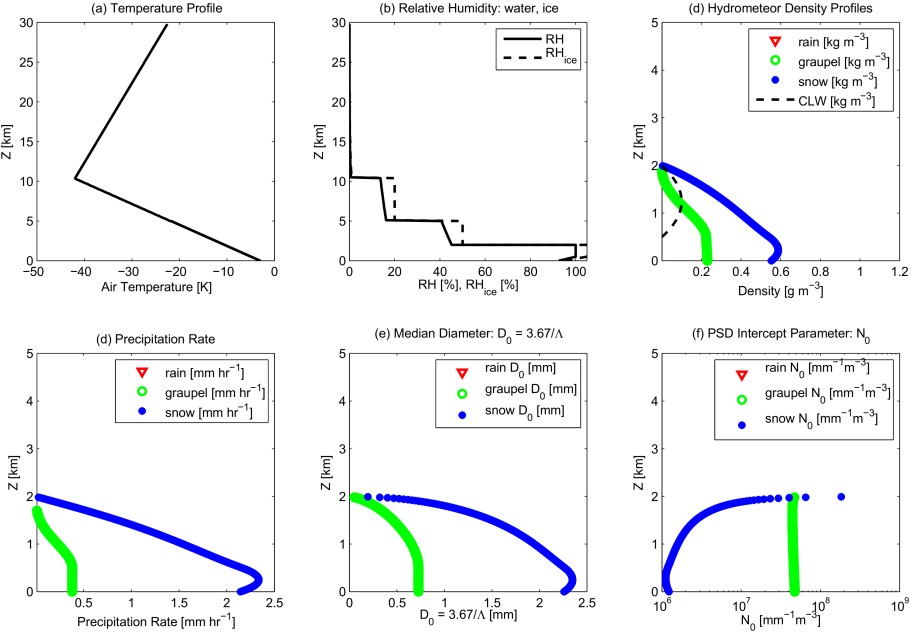

**Figure 2.** Example physical properties and associated PSD properties used during stand-alone testing of CASM. (a) vertical profile of temperature, (b) relative humidity (%), (c) hydrometeor density, (d) liquid equivalent precipitation rate for graupel and snow, (e) median diameter D0 of the exponential particle size distribution, and (f) the intercept parameter, N0 of the exponential size distribution.

The dielectric constant for ice is from the tabulation of Warren and Brandt (2008), and the dielectric constant for liquid water comes from Liebe et al. (1991). For the dielectric mixture of ice, water, and air (as needed), we use the Bruggeman method. For a more detailed description, see Johnson (2007); Johnson et al. (2012).

The Mie scattering model, developed partially by K. F. Evans was included as part of the "RT4" package (Evans and Stephens, 2014), and has been heavily modified by the author to be more flexible and extensible to non-spherical particles. The primary inputs are temperature, dielectric constant (averaged), PSD slope and intercept parameters, and wavelength of microwave radiation. Outputs are the vertically and horizontally polarized coefficients of extinction, scattering, and radar backscattering. Also produced is the scattering asymmetry parameter (degree of forward or backward scattering), and the full Stokes scattering phase function, stored as coefficients of the Legendre expansion of the phase function at specific quadrature angles (typically 11 angles, including 180 degrees for backscattering calculations).

These extinction and scattering properties are integrated over the particle size distribution within the Mie model. Hereafter, references to any of the above properties imply that they are the PSD-integrated quantities, unless otherwise specified.





Figure 2 shows an example of the PSD-integrated optical properties for the profile given in Fig. 2 at 13.4 GHz (Ku-band) and 35.6 GHz (Ka-band) – both consistent with the Global Precipitation Measurement mission (GPM) Dual-frequency Precipitation Radar (DPR).

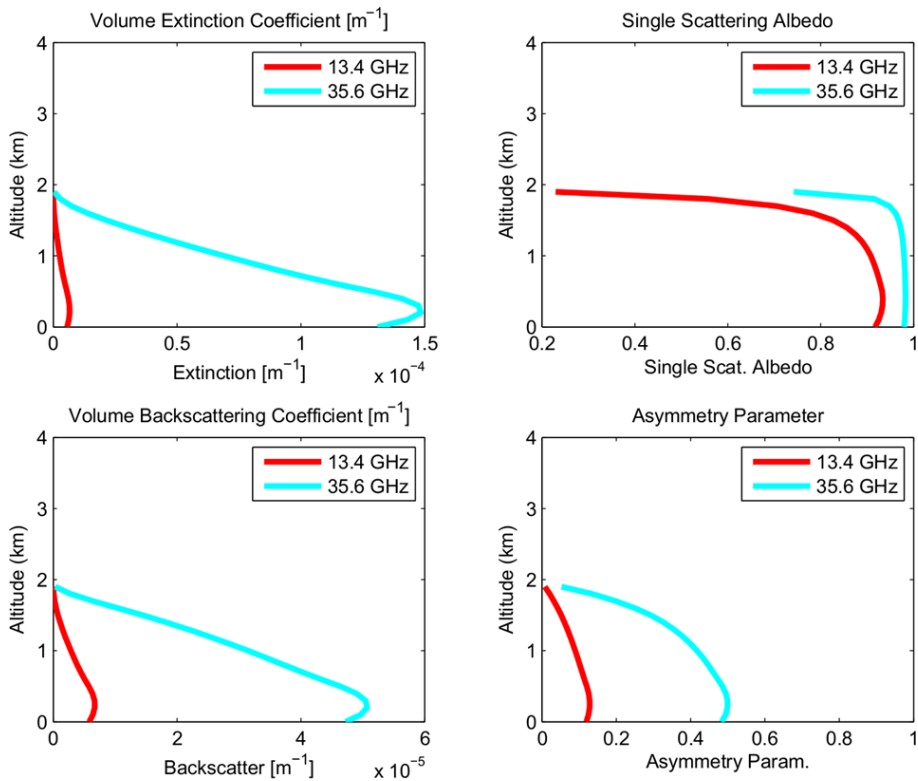

**Figure 3.** Example PSD-integrated scattering and extinction properties for the physical profiles shown in Fig. 2.

## 2.2 Simulation of Atmospheric Reflectivity and Path-Integrated Attenuation

Given the PSD-integrated optical properties, the profile of extinction, scattering, and backscattering can be translated into observable quantities using an appropriate radiative transfer model. In this case, a modified version of the RT4 package includes an adding-doubling model for simulating the thermal emission and upwelling of microwave radiation through a 1-D atmosphere. The top-of-the atmosphere passive microwave brightness temperatures are computed, and can be compared with observations or other models. Furthermore, at each layer, the radar reflectivities for that layer are computed using the following relationship:

$$Z_{\text{eff}} = 10 \log_{10} \left( 10^{18} \frac{\lambda^4}{|k^2| \pi^5} C_{\text{back}} \right) \tag{2}$$

with units





$$[\text{dBZ}] = 10\log_{10}\left([10^{18}\text{mm}^6\text{m}^{-3}][\text{m}^4][\text{m}^{-1}]\right)$$

where $C_{\text{back}}$ is the radar backscattering cross-section, computed using Mie theory in this study. Unfortunately, the standard CRTM scattering/extinction database does not explicitly provide the backscattering information needed. Instead, the scattering phase function is provided via coefficients from the Legendre expansion for the given hydrometeor category, and the backscat-

5 tering is estimated from the phase function information, using an appropriate amplitude scaling.

## 2.3 CRTM Libraries and Modifications Used in CASM

CASM requires an external model to describe the single particle scattering and extinction properties of precipitation hydrometeors. It also does not prescribe a specific particle size distribution (PSD), and relies on external libraries to provide this information. In this work, the hydrometeor model from CRTM is used.

The hydrometeor model in CRTM utilizes a static binary file as a look-up table (LUT), providing the necessary extinction, scattering, and asymmetry parameter information when provided with a frequency, effective radius (radius of an equal-volume sphere), and temperature (in the case of rainfall only). Specific care has been taken to relate the slope parameter of the exponential particle size distribution to either radius or diameter, as appropriate – these terms are used interchangeably. Within the CRTM hydrometeor library, the scattering LUT contains information for both microwave and infrared wavelengths, and for

liquid and solid particles. There is one class of liquid-phase hydrometeors and four classes of ice-phase hydrometeors: snow, graupel, hail, and cloud ice – each with a pre-defined bulk density.

The outputs of the CRTM hydrometeor model are PSD-integrated mass extinction coefficient (mass weighted extinction cross-section, $\text{m}^2\text{kg}^{-1}$), scattering asymmetry parameter (ranging from -1 to 1), single scattering albedo (ratio of scattering to total extinction, ranging from 0 to 1), and the Legendre coefficients of the scattering phase function, which have up to 38 terms

(amplitude only, no polarization). Of note is the lack of a radar backscattering cross-section in the look-up table, nor is this necessary quantity currently computed by default in CRTM libraries.

Given this limitation, we sought to roughly estimate the backscattering cross-section given the total scattering cross-section and the intensity value of the scattering phase function at 180 degrees (i.e., "backward" scattering.) Starting with the scattering phase function, $p(\tau, \Theta)$, as a function of the optical path $\tau$, and the scattering angle $\Theta$, it is expressed as the sum of the Legendre

coefficients, $P_n(\cos\Theta)$, and the amplitude weights $(\chi_n(\tau))$ as follows,

$$p(\tau, \Theta) \approx \sum_{n=0}^{2N-1} (2n+1)\chi_n(\tau)P_n(\cos\Theta). \tag{3}$$

At $\Theta = 180$ degrees, the Legendre expansion coefficients are

$$P_n(-1) = (1, -1, 1, -1, ...). \tag{4}$$





The scattering cross-section, $C_{\text{scat}}$, is a product of the provided mass-extinction coefficient $M_{\text{ext}}$, the single scattering albedo, $\omega$, and the cross-sectional area of the particle $A$. To obtain the layer-averaged scattering cross-section, this is scaled by the geometric layer thickness $\delta z$ and layer water content $W$:

$$C_{\text{scat}} = \frac{W}{\delta z} M_{\text{ext}}(D)\,\omega(D)\,A(D).$$ (5)

The layer-averaged backscattering cross-section is, consequently, the product of the PSD-averaged phase function (eqs. 3 and 4) and the layer-averaged scattering cross-section (see Bohren and Huffman (1983) for a detailed discussion).

$$C_{\text{back}} = C_{\text{scat}}\,p(\tau, \Theta = 180°)$$ (6)

Provided with a vertical profile of water content and a particular hydrometeor type, the layer-averaged radar backscattering coefficient and the layer-averaged extinction can be computed for a layer of hydrometeors, for each hydrometeor type present. From the hydrometeor scattering and extinction perspective, there are no constraints between adjacent layers (i.e., each layer is treated as physically independent from the adjacent layers).

The two-way path-integrated attenuation (PIA) assumes a cloud top-down integration approach, where attenuation "accumulates" moving down through the cloud. It also assumes that the radar signal is attenuated in the same manner on the return trip. Three primary contributors to the path-integrated attenuation are considered: (1) absorption by gases, particularly water vapor and "air", (2) absorption by cloud liquid water, and (3) absorption + scattering by hydrometeors as the primary contributor.

Like the backscattering cross-section, the two-way path integrated attenuation is computed using the total extinction provided from the CRTM LUT at each layer, and the gaseous extinction provided by ancillary observations. The PIA is written as follows:

$$\mathcal{A}(z) = \exp\left(-2\int_{r_0}^{r} (k_{\text{scat}}(z) + k_{\text{abs}})\,\mathrm{d}z\right),$$ (7)

where $r_0$ is the geometric distance to the first range gate from the radar, and $r$ is the distance from $r_0$ to the current range gate, and $k_{\text{scat}}$ and $k_{\text{abs}}$ are the unitless volume scattering and absorption coefficients, where $k_* = C_*/A$. The attenuation is then multiplied by $Z_{eff}$ (prior to dBZ conversion) to obtain the attenuated reflectivity ($Z_{\text{m}}$) – consistent with what would be observed by a satellite or aircraft radar. This simulated attenuated reflectivity can be directly compared to typical observed radar reflectivities, after conversion to dBZ. In the following sections, the term "corrected" reflectivity refers to the simulation of reflectivities without the attenuation contribution. In the case of radar observations, this correction is usually provided in the data product.

## 2.4 Reference Model Comparison

In order to assess the range of validity of CASM, the reference model Johnson (2007); Johnson et al. (2012) was developed to simulate the range of applicability of Mie Spheres and computed backscattering cross-sections (see section 2.1).



Given a profile of PSD-integrated radar backscattering coefficients, vertical profiles of radar reflectivities are produced using CASM. Figures 4–6 show examples of how the CASM reflectivities (black lines) compare to the stand-alone reference model.

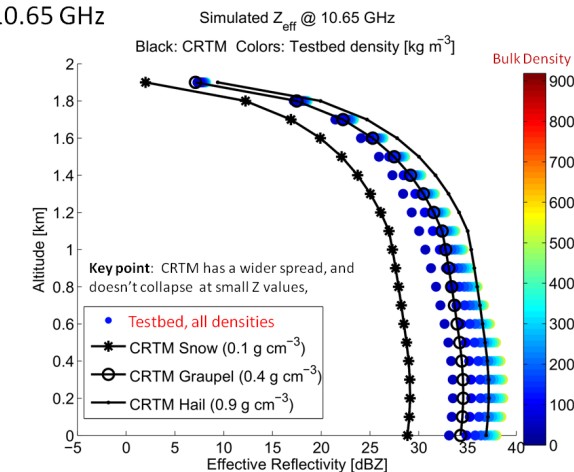

**Figure 4.** Comparison of CASM computed reflectivities at 10.65 GHz using the default CRTM scattering database (black lines) compared to the reference model (described in the text) as a function of bulk hydrometeor density (colored points). For consistency, both scattering databases the bulk particle density is the same for all sizes in the integrated particle size distribution.

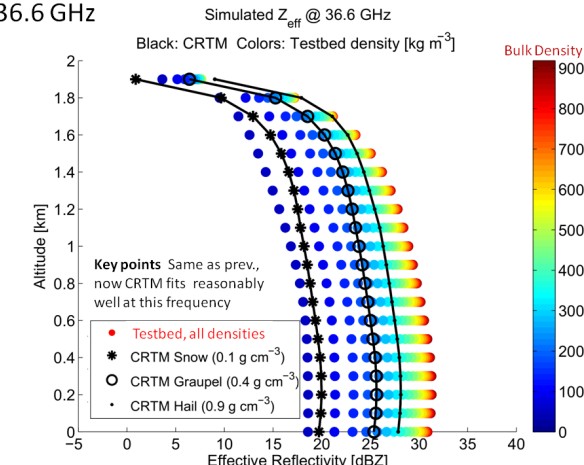

**Figure 5.** Same as figure 4 except at a frequency of 36.6 GHz.

To generate these profiles (black lines), CASM was provided with the same vertical profile of snow water contents as was used in the reference model, and simulated using the same effective radius at each frequency. The densities in the reference model span the range from $0.1 \, \mathrm{g \, cm^{-3}}$ (dark blue) to $0.9 \, \mathrm{g \, cm^{-3}}$ (dark red). In all cases, in spite of the coarse nature of





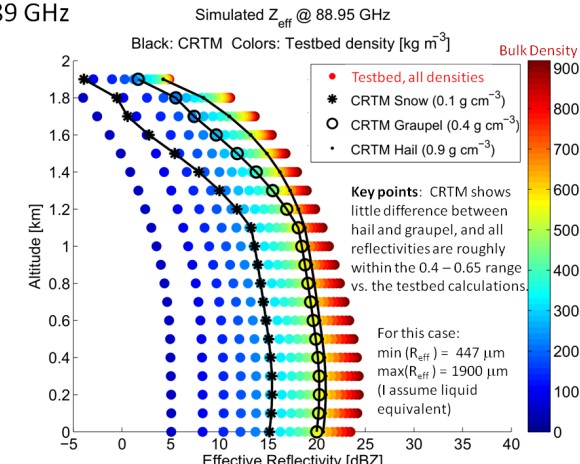

**Figure 6.** Same as figure 4 except at a frequency of 89.0 GHz.

the approximate estimate of the backscattering coefficient, CASM reproduces roughly the same shape and magnitudes of the reflectivities compared to the reference model.

Examination of the 89 GHz fig. 6 shows a breakpoint for reflectivities around an altitude of 1.5 km, where the reflectivities of the hail and graupel density particles start to converge with increasing reflectivity. Figure 7 below illustrates the issue more clearly by plotting the simulated reflectivities versus the effective radius.

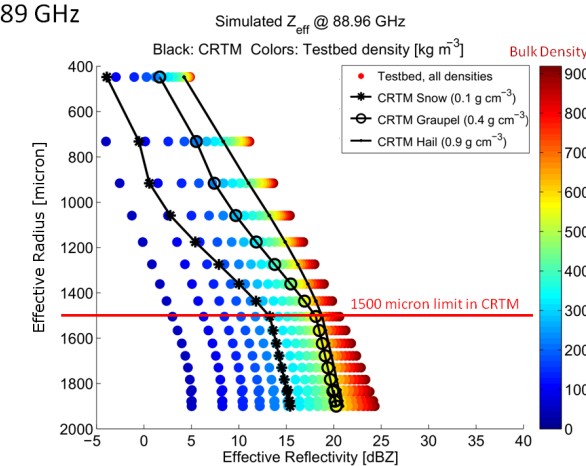

**Figure 7.** Effects of CRTM's 1500 micron effective radius limit on computed reflectivities.

The breakpoint in Fig. 7 occurs at 1500 microns effective radius (vertical axis), which is also the maximum limit of effective radius in the default CRTM scattering database. This highlights a serious limitation of the existing database in CRTM. Development is underway for an extended version of the database to allow for the accurate simulation of reflectivities (and all





other parameters) at larger effective radii, and over a much wider range of microwave frequencies to support current and future satellite data assimilation efforts.

## 3  Simulation Studies and Validation

To validate the performance of CASM, GPM DPR radar reflectivity and attenuation observations are compared with CASM
simulations. In the first section, a case study containing a variety of precipitation types (rain, mixed-phase, snowfall), ranging from light to heavy precipitation was selected for comparison. The second section compares CASM simulations against a year-long global dataset of DPR observations at Ku and Ka-band.

### 3.1  Case Study Comparison of CASM and GPM DPR Observations

Level 2A GPM DPR data files were obtained from the official GPM data server. There is a separate data file for Ku- and Ka-
bands. Within each file, a number of parameters are present: For the present validation, the Ku-band reflectivity and attenuation measurements were used as observational data, and the derived mass-weighted median diameter ($D_\mathrm{m}$) and intercept number concentration ($N_\mathrm{w}$) were derived using the surface reference technique method described in Seto and Iguchi (2015). Using these two parameters, CASM (with the CRTM scattering library) was used to compute the effective radar reflectivity and path-integrated attenuation, as described in previous sections. Figure 8 shows a 2-D slice through the 3-D volume of these two
parameters. In the melting layer region, odd behavior of $D_\mathrm{m}$ and $N_\mathrm{w}$ is evident – this is primarily due to a lack of an explicit melting layer model in the official GPM-DPR level 2 retrieval algorithm, and it compensates by adjusting the PSD parameters to force a fit to the reflectivities.

Figure 9 shows that CASM can accurately reproduce single-profile Ku-band reflectivities with high fidelity, with the notable exception of the melting layer region, where both CASM and level 2A algorithm are lacking an explicit melting layer model.
Extending this single-profile to the 2-D slice is shown in figure 10.

This early version of CASM is performing well for cases where the vertical profile is continuous, but appears to be suffering in regions where the reflectivity column is broken or marginal (e.g., the right-hand side of Fig. 10 (c) and (d)). Further investigation into these artifacts is required, and will be a part of the next round of updates to CASM.

### 3.2  Validation Against a Global DPR Dataset

Extending the comparison above to global dataset allows for a more robust statistical comparison. One year of GPM DPR level 2A (V6) data was downloaded and processed. Following the approach above, the $N_\mathrm{w}$, $D_\mathrm{m}$, temperature, and observed reflectivity was obtained from each DPR file. To avoid sidelobe clutter effects (Furukawa et al., 2013), only the nadir beam was selected. Processing these variables using CASM, figure 11 shows CFADs (Contoured Frequency by Altitude Diagrams) of attenuation-corrected reflectivity ($Z_c$) for GPM DPR observed Ku- and Ka-band reflectivities in panels (a) and (b), respectively;
and for the CASM simulations at Ku- and Ka-band in panels (c) and (d), respectively.





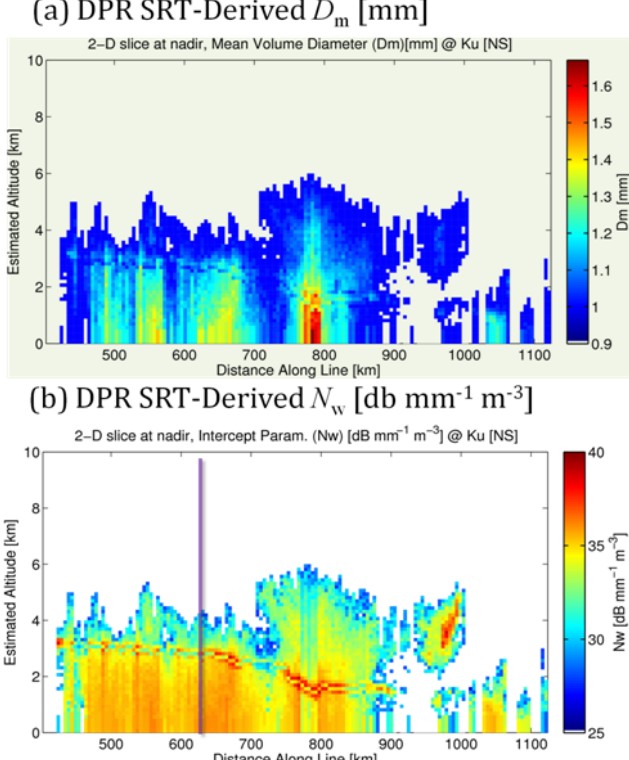

**Figure 8.** Particle size distribution parameters $D_m$ and $N_w$ derived from GPM DPR Ku-band observations. These parameters are used in CASM to forward model the reflectivities and compute the attenuation.

To facilitate comparisons between different profiles, altitudes were normalized as "height above the surface". Range bin altitude is not specifically provided in the level 2A GPM-DPR product, it was estimated using the number of range gates in the Ku-band (NS) products and the Ka-band (MS) products. The Ku- and Ka-band for the matched scans (NS and MS) have the same range resolution (250 meters). Radar echoes are over-sampled at twice the rate of the corresponding resolution, i.e.,125 meters for the matched beams. CASM simulated reflectivities were computed at the same range gates as the DPR observations, resulting in a one-to-one correspondence between observation and simulation on a profile-by-profile basis. In fig. 11, the vertical bin resolution was set to 250 meters to provide visually smoother sampling, *but no modifications were made to either the simulated or observed data itself to improve the fit*. Similarly, the reflectivity bins are set to 1 dBZ width.

In fig. 11 there's an apparent bias in the Ku-band CASM simulations, and a low-end cutoff around 14 dBZ. The DPR observations go down to around 12 dBZ (the minimum detectable threshhold). This cutoff in CASM simulations may be artificial. Otherwise the structure of the CASM Ku-band CFAD looks consistent with the observations, in both intensity and spread. On the Ka-band side (panels (b) and (d)), the comparison looks cleaner in the core reflectivity regions, but CASM appears to be underestimating both higher altitude precipitation and low-level high-reflectivity precipitation in the 30-36 dBZ range. There is also a significant overestimate of reflectivity at the highest values (40+ dBZ) – the cause of this appears to be





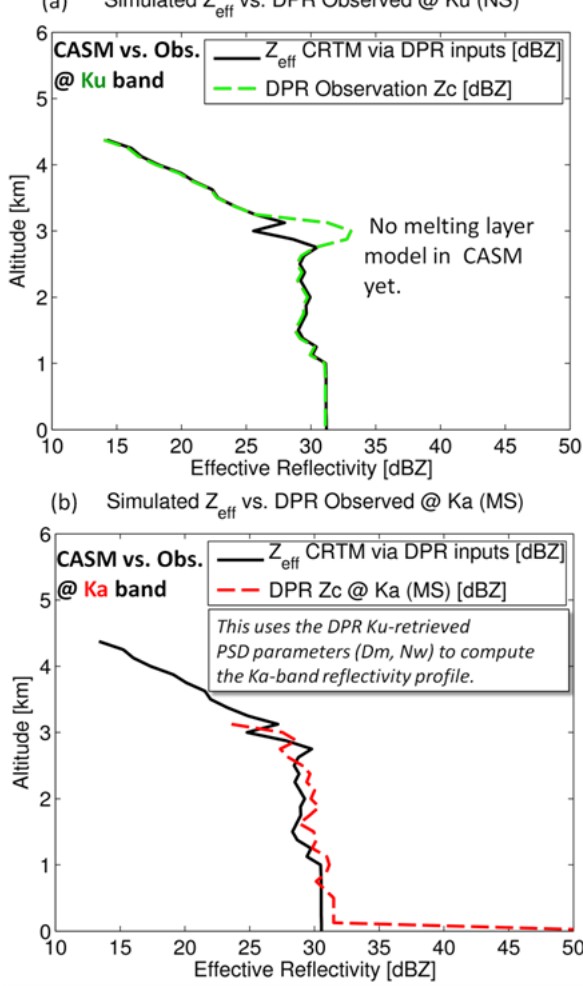

**Figure 9.** Simulated reflectivity using CASM (black line) compared to observed reflectivity (dashed lines), for Ku-band (panel a) and Ka-band (panel b). For Ka-band, the derived PSD parameters from Ku-band were used.

unexplained shifts in the estimated $D_m$ and $N_w$ profiles. Examination of individual contiguous observed reflectivity profiles, shows occasional dramatic departures of $D_m$ and $N_w$ from nearby similar profiles. This is believed to be a feature of the DPR level 2A processing algorithm, and is not under the control of the author.

## 4   Conclusions

5   CASM, using CRTM libraries, produces vertical profiles of radar reflectivity and path-integrated attenuation. Given the noted limitations of the CRTM scattering lookup table, particularly the maximum effective radius at 1500 microns, we find that radar reflectivity simulations suffer in cases of heavy precipitation where the effective radius exceeds this limit. Comparisons against





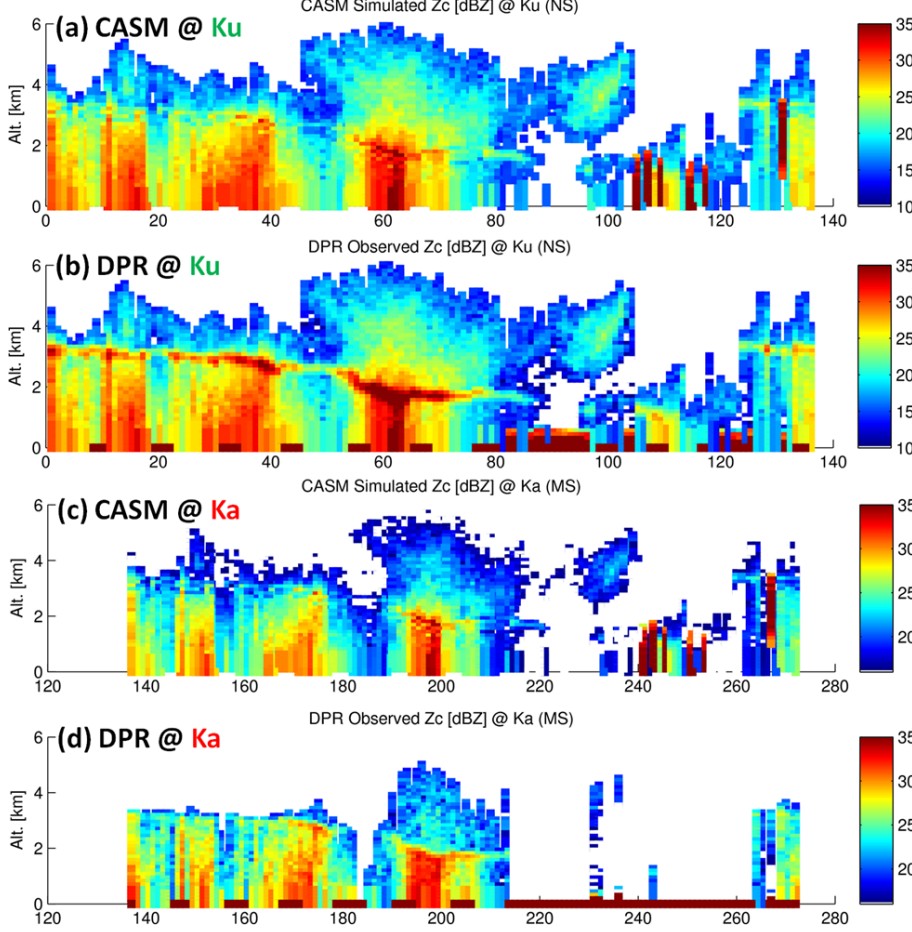

**Figure 10.** Attenuation-corrected radar reflectivities: (a) is the CASM simulation at Ku-band, (b) is DPR corrected reflectivities at Ku-band. (c) and (d) are the same as (a) and (b), except at Ka-band. Note the significant attenuation corrections at Ka band,

GPM observations show a remarkable ability to accurately reproduce the observations using the DPR-provided $D_m$ and $N_w$ parameters.

Future research will explore the integration of CASM into MIIDAPS, starting with the computation of the Jacobians of active sensor observables, which will allow for accurate derivation of atmospheric and surface parameters in an analysis framework
5  (e.g., wind, wave height, cloud and precipitation profiles, etc.). Ultimately CASM is expected to provide a full active microwave sensor simulation capability, for all-weather and all-surface conditions. The tangent-linear and adjoint components of CASM, and subsequent Jacobian calculations, provides the capability of directly interfacing with current numerical weather prediction analysis packages, such as the Global Data Assimilation System (GDAS) at NOAA – an integral component of the operational weather prediction capability in the U.S.





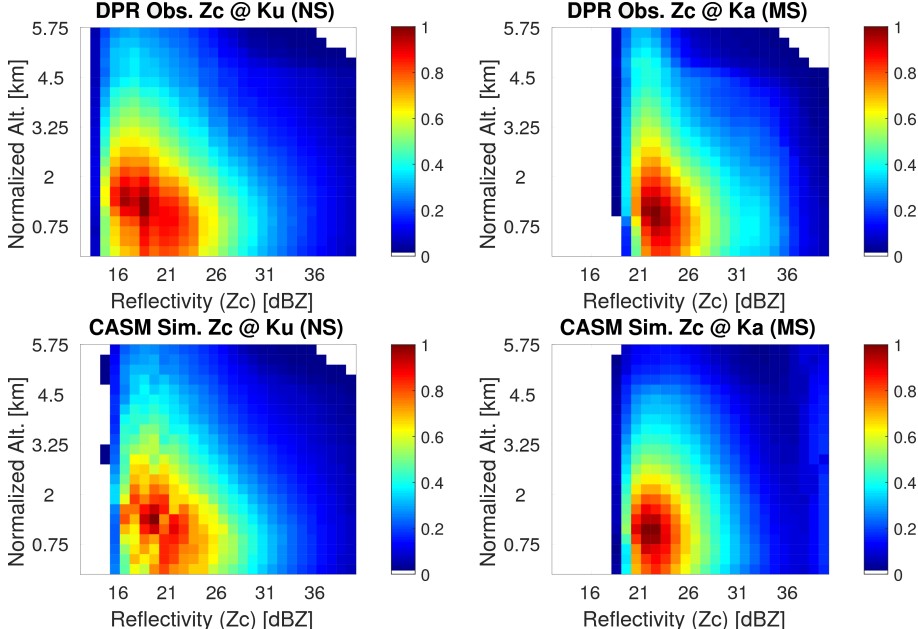

**Figure 11.** CFADs of GPM DPR observations (top row) and CASM simulations (bottom row) for Ku band and Ka band (beam matched). One year of data was used for the analysis, at nadir beam only, from 01 January 2015 to 31 December 2015. Approximately 4 million reflectivity profiles at both Ku- and Ka-band were used in the analysis.

*Acknowledgements.*



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
