# Peer review of "Development of the Community Active Sensor Module (CASM): Forward Simulation"

_Atmospheric Measurement Techniques, 2016_

## Referee Comment (RC1) · Anonymous Referee #1 · 22 Jul 2016

**General Comments**

The manuscript "Development of the Community Active Sensor Module (CASM): Forward simulation" describes the development and validation of a radar forward operator designed for use in the Community Radiative Transfer Model (CRTM), which is used for both retrievals and data assimilation. While the development of such an operator is not novel, the application of such an operator in an assimilation framework is an important step in utilizing active remote sensing observations in numerical weather prediction. The manuscript is detailed, straight-forward, and well-written; however, there are some details of the manuscript that should be discussed and addressed before publication.

**Specific Comments for Discussion**

[Figure]

On page 6, lines 10–16, the authors discuss the specifics of the hydrometeor model LUT. While some of this information is available in other places, a table giving the particle densities (for frozen hydrometeors), the temperature range (for liquid), and the effective radius/diameter ranges would be helpful.

The authors use Figure 6 to address the effective diameter limit (of 1500 $\mu$m) in the scattering tables; however, it seems as though the effects are also visible in Figure 5, where $Z_{eff}$, especially for the hail case, starts to diverge from the reference model. This can be seen to a lesser degree in Figure 4 for lower altitudes. The authors should expand this discussion cover all three figures, instead of just Figure 6.

On page 10, lines 15–17, the authors state that an explicit melting model is not used in the GPM-DPR level 2 algorithm. Seto et al. (2013), which describes the algorithm, includes the characterization of the melting layer, both for the effective permittivity, and for the modification of the PSD. This procedure is similar to what was described by Iguchi et al. (2000) for the single frequency retrievals from TRMM-PR.

In Figure 10 there are some high reflectivity artifacts near the surface, but these are not addressed in the manuscript.

The CFADs constructed to analyze the full-year comparison are of attenuation-corrected reflectivity. It seems as this would remove the attenuation component of the modeling, while also introducing some error into the measurements due to the GPM attenuation correction algorithm (I acknowledge that these errors are propagated into the profiles that are used as input to the forward operator). It would be nice to also see comparisons of the uncorrected reflectivities. Additionally, the figure might be more instructive if it gave relative frequency instead of normalized frequency in the colorbars.

In the conclusions, the tangent-linear, and adjoint components are discussed as though these capabilities are already included, even though they are planned enhancements. The text should be modified to make this clear.

**Technical Corrections**

Page 2, line 11: "combined CRTM" should be "combined with CRTM".

Page 2, line 13: There is a closing parenthesis without a corresponding opening parenthesis.

Page 3, Figure 2: A symbol is given for rain, but that symbol does not appear in the actual plotted data. The symbol should be removed from the legend.

Page 2, line 6: A comma should be placed between "Evans" and "was."

Page 5, line 2: Except for $C_{back}$, none of the terms of the equation are defined.

Pages 8-9, Figures 5–7: The densities have different units in the caption and for the colorbar. Use consistent units.

Page 8, line 3: remove the "s" from the end of "contents."

Page 10, line 9: Here the Level 2A GPM DPR product is introduced, but there is no data citation. Please include a data product citation in the references.

Page 11, lines 7–8: The italics are unnecessary.

Page 13, lines 6–7 For "The tangent-linear and adjoint components of CASM, and subsequent Jacobian calculations, provides..." is incorrect in number (i.e., should be "provide").

Page 14, Figure 11: The colorbar should be defined in the caption.

**References**

Iguchi, T., Kozu, T., Meneghini, R., Awaka, J., and Okamoto, K.: Rain-Profiling Algorithm for the TRMM Precipitation Radar. *J. Appl. Meteor.*, 39, 12, 2038—2052, 2000.

Seto, S., Iguchi, T., and Oki, T.: The Basic Performance of a Precipitation Retrieval Algorithm for the Global Precipitation Measurement Mission's Single/Dual-Frequency

Radar Measurements, *IEEE Trans. Geosci. Remote Sens.*, 51, 12, 5239–5251, 2013.

---

## Referee Comment (RC2) · Anonymous Referee #2 · 29 Jul 2016

The paper presents a forward model active radar simulator that is potentially useful in data assimilation. After the technical description of the simulator, the outputs of the simulators are compared with GPM observations. I seriously struggle in finding the novelty in this paper. 1) The simulator itself (in its current version) is the simplest possible I could imagine: just nadir, no 3D, no antenna pattern, no Doppler, no surface modelling, no multiple scattering. Current state of the art radar simulators are far more advanced (and no reference is actually provided to all the advances made in the past 10 years on this topic, e.g. see works by Tanelli, Kollias, Battaglia, Hogan and I am probably forgetting many others in preparation of future radar missions like EarthCARE, ACE). I do not see any advance even with respect to Quikbeam (paper

published almost 10 years ago) or to the simulator developed by R. Hogan. If the simulator is simpler because it wants to be extremely fast and capable of computing the adjoint and Jacobians this should be demonstrated (but is left as future work). Moreover if you claim that it is useful for altimeters and scatterometers you should show examples where you can simulate the surface returns (as seen by altimeters!). 2) The other serious problem I see is represented by the scattering libraries. How is possible to only include Mie spheres (in the current version) and only particles with sizes smaller than 1.5mm? All LUT can be very quickly updated. 3) All the description in Sect2.3 is pretty contorted and confused. Computing single scattering scattering and backscattering cross sections is straightforward. Integrating over PSD provides then extinction, scattering and backscattering coefficients (details are found in text books, no need to repeat them). Formula (6) is indeed exact (the backscattering cross section by definition is the scattering cross section at Theta=180) and not a rough approximation. A lot of imprecise statements (e.g. What is "air" (dry air)? The attenuation indeed starts from 0 not r0, kscat and kext are NOT UNITLESS but L^-1 , confusion in formula (5)). 4) I do not understand what you are comparing here. If you are using Mie spheres with the same mixing rule all results should be spot on. Where is the difference coming from? 5) Sect.3. Again I do not see what you want to prove here. Essentially you are comparing your model with the GPM forward model and LUT. If you are using the same LUT as GPM your results should be spot on (not really because the retrieved profiles are not perfectly reproducing the observations but if I remember well there are forward modelled reflectivities in GPM files). The only thing that you are indeed proving is that you have some problem/bug in your code as clearly highlighted by what is happening in Fig.10 around pixels 100-120. Given all these weaknesses I deem the article not suited for publication and I would review it again only if my criticisms are seriously addressed. I do not want to demoralize the authors; their work is indeed potentially very useful but a lot more must be done to level their work with the state of the art in this research field.